# Examination of Physical Characteristics and Positional Differences in Professional Soccer Players in Qatar

**DOI:** 10.3390/sports7010009

**Published:** 2018-12-31

**Authors:** Eirik Halvorsen Wik, Seán Mc Auliffe, Paul James Read

**Affiliations:** 1Aspetar Sports Injury and Illness Prevention Programme (ASPREV), Aspetar Orthopaedic and Sports Medicine Hospital, P.O. Box 29222, Doha, Qatar; 2Athlete Health and Performance Research Centre (AHP), Aspetar Orthopaedic and Sports Medicine Hospital, P.O. Box 29222, Doha, Qatar; sean.auliffe@aspetar.com (S.M.A.); paul.read@aspetar.com (P.J.R.)

**Keywords:** football, Arabic, strength, jump, range of motion, movement screening, player profiling

## Abstract

Physical characteristics in professional soccer differ between competition levels and playing positions, and normative data aid practitioners in profiling their players to optimize performance and reduce injury risk. Given the paucity of research in Arabic soccer populations, the purpose of this study was to provide position-specific normative values for professional players competing in the Qatar Stars League. One hundred and ninety-five players completed a musculoskeletal assessment as part of an annual periodic health examination. Tests included measures of range of motion (hip, ankle, and hamstring), bilateral and unilateral jump performance, and quadriceps/hamstring (isokinetic/NordBord), hip adduction/abduction (eccentric), and groin (isometric) strength. Descriptive data were examined, and positional differences were analyzed using a one-way analysis of variance (ANOVA). Goalkeepers were significantly heavier (*p* < 0.01), had a higher body mass index (*p* < 0.05) than outfield positions and demonstrated greater absolute strength. Defenders were the strongest relative to body mass, and these differences were significant (*p* < 0.05) versus goalkeepers and strikers. No meaningful between-group comparisons were apparent for jumping or range of motion tests. Compared to mean values from other professional leagues, soccer players in Qatar appear to be shorter, lighter and display inferior strength and jump capacities. These data can be used to tailor training and rehabilitation programs to the specifics of the league and position in which the athletes compete.

## 1. Introduction

Soccer performance is a complex concept, with multiple factors interacting, including physiological, psychological, tactical, and technical aspects. Physical demands vary between playing positions [1] and performance levels [2], and assessments of physical characteristics have, therefore, been used for talent identification [3], individual fitness profiling [4], and to identify injury risk factors [5]. Consequently, it is of importance that coaches and clinicians can assess and evaluate the physical status of a player or team based on comparable context and position-specific normative data. To date, several studies have investigated the physical characteristics of players in elite European leagues [4,6,7,8,9]; however, data to describe elite soccer players in the Arabian Gulf states are sparse. 

Equivocal evidence is available to show a reduced range of motion (ROM) or strength as injury risk factors [10,11,12,13,14,15], while maximal strength and jump performance has been correlated to sprint performance [16], which again is considered important in match decisive playing situations [17]. Although recent studies have suggested that a single musculoskeletal screening test cannot predict which professional soccer players will go on to sustain a future injury [18,19], these tests can provide value in terms of detecting current problems and establishing non-injured baseline values [20]. Including these tests as part of a pre-participation evaluation could, therefore, provide practitioners with relevant performance indications that can guide training and rehabilitation programs.

Aerobic, anaerobic, and anthropometric characteristics have been reported in previous literature for professional players in the United Arab Emirates (UAE) [21]. Although position-specific values were provided, the study was limited to a selection of physiological characteristics, while excluding strength, range of motion, and jump performance test variables. Physiological profiles have also been reported for national team players of Saudi Arabia [22] and Kuwait [23]; however, these studies do not report position-specific or pre-participation musculoskeletal screening data and were performed more than two decades ago. Information on anthropometric values can also be found for elite soccer players in Kuwait [24] and Bahrain [25], although the former study included a considerably small sample with measures used as baseline values to examine the effects of fluid loss following soccer match play and the latter is, again, more than two decades old and does not include measures of physical function.

In Qatar, normative values for professional soccer players have been reported for hip strength and range of motion [26], and strength and flexibility have been examined in studies assessing injury risk factors [5,18,19,27]. In these studies, the authors did not report or compare position-specific values, nor did they include measures of jump performance. Normative position-specific strength and range of motion values have important implications for injury management and rehabilitation. Ascertaining whether an athlete has regained muscular function or range of motion compared to the non-injured side is a commonly used outcome of rehabilitation; however, investigating whether the athlete has returned to baseline values is also an important component of rehabilitation outcomes. Measures of jump performance are routinely used to monitor physical performance and fatigue [28,29] and have recently been included as part of pre-participation injury risk screening in elite youth soccer players [30].

Establishing population and position-specific normative performance values will allow practitioners to more accurately identify deficiencies in performance that are relative to their athletes. Given the ever-growing popularity of soccer in the Arabian Gulf states, in conjunction with the upcoming FIFA World Cup in Qatar 2022, the need for a comprehensive profile of the physical characteristics displayed by players competing at the elite level within this particular region is highlighted. This study aimed to examine positional differences in commonly used clinical and performance tests in players participating in the Qatar Stars League (QSL).

## 2. Materials and Methods

### 2.1. Experimental Design

Professional soccer players from the QSL attended Aspetar Orthopaedic and Sports Medicine Hospital in Doha, Qatar, during the pre-season or early competition period of the 2017–2018 season, as part of a FIFA-compliant annual periodic health evaluation (PHE). The PHE involved laboratory blood analysis, dental and cardiac examination amongst others. In addition, players engaged in a musculoskeletal screening assessment which consisted of range of motion, strength, and dynamic screening tests.

### 2.2. Participants

One hundred and ninety-five professional soccer players registered at clubs competing in the QSL were recruited to participate in this study. The QSL is the highest level of professional soccer in Qatar, consisting of twelve teams, where the top three clubs qualify for either the group stage (winner) or play-off rounds (second and third) of the Asian Football Confederation (AFC) Champions League. Demographic information is displayed in Table 1. Inclusion criteria required players to be registered as a professional soccer player in the QSL, free from current injury at the time of testing, and currently participating in full soccer training. All participants provided informed consent before data collection, and ethical approval was obtained from the Anti-Doping Lab Qatar Institutional Review Board (IRB: E2013000003). For players under 18 years, informed consent was obtained from parents or guardians.

### 2.3. Procedures

Participants were requested to eat according to their normal diet in the day preceding the assessment and then refrain from eating and drinking substances other than water one hour before. Upon arrival, participants were provided with appropriate explanation and demonstration of all procedures. Additionally, athlete-reported playing position and leg dominance were recorded. Positional groupings were goalkeepers (GK), defenders (DEF), midfielders (MID) and strikers (STR), and the dominant leg was defined as their preferred kicking leg for a penalty kick. Anthropometric information was recorded before players completed a standardized warm-up consisting of 10 min of cycle ergometry. For all strength and jump tests, warm-up trials were provided to familiarize each participant and to ensure technical competence during the test. The tests were performed in a standardized order, with jump performance tests preceding strength measures.

#### 2.3.1. Range of Motion

Hip range of motion was measured using the bent knee fall out (BKFO) test and internal rotation (IR) in a position of 90° of hip flexion as described previously [10,26,31]. BKFO was measured as the distance between the most distal point on the head of the fibula and the surface of the plinth using a tape measure, while IR in 90° hip flexion was measured using a manual goniometer. A single test was used for the BKFO distance, and three repetitions were used for hip IR with the average score recorded.

To determine hamstring flexibility, a passive knee extension test (PKET) was performed as previously described [32]. The athlete positioned their hip in maximal flexion by clutching the thigh to their chest, and the contralateral leg was fixed in place by the assessor. The maximal angle was measured using a handheld inclinometer (Digital Level Box JY-180ALB, Grand Index, Hong Kong) positioned on the tibia.

Ankle dorsiflexion was measured using the weight-bearing lunge test with the athlete facing a wall in a standing position [33]. Range of motion was recorded using a tape measure and determined as the maximum distance away from the wall that the knee of the lead (testing leg) was able to touch during the forward lunge. Participants were instructed to maintain heel contact with the ground throughout the test. All testing was completed with the participants barefoot, and measures were recorded to the nearest 0.1 cm.

#### 2.3.2. Jump Performance

Jump performance was assessed through countermovement jumps and a single-leg 10-s Hop test. For countermovement jumps, participants began either in a bilateral (CMJ) or a unilateral (SLCMJ) stance on a force plate (Force Decks v1.2.6109, Vald Performance, Albion, Australia) with their hands on hips and opposite hip flexed at 90°. Instructions were to perform a countermovement by dropping into a quarter squat and then immediately triple extending at the ankle, knee, and hip in an explosive concentric action. Bending of the knees while airborne was not permitted, and participants were required to repeat the test if this occurred. Jump height was calculated from the athlete’s flight time and recorded to the nearest 0.1 cm. All data were recorded at a sampling rate of 1000 Hz. Three trials were completed with a 30 s rest period between each repetition. 

For the single-leg 10 s Hop test (10 s Hop), a series of repeated single-leg jumps were performed for a period of 10 s in accordance with previous guidelines [34]. The test began with participants completing a rapid countermovement into a quarter squat followed by a maximal vertical jump. This action was replicated across the test capture period. Instructions were to jump as high as possible, minimize ground contact time while landing under control and try to maintain the same footprint, remaining facing forwards during the entirety of the test. One trial was completed on each leg. An optical measurement device (Optojump, Micrograte, Bolzano, Italy) was used to quantify average jump height and the reactive strength index (RSI) derived from flight time divided by ground contact time [35].

#### 2.3.3. Strength

Maximal strength was measured using isokinetic tests, the Nordic hamstring exercise and hand-held dynamometers. Quadriceps (knee extension) and hamstring (knee flexion) strength profiles were measured using an isokinetic dynamometer (Biodex Medical Systems, Shirley, New York, USA). Players were seated in a position with the hip flexed to 90° with all procedures replicating those of previous research [5]. Assessment modes included five repetitions of concentric knee flexion and extension at 60 deg/s (QCon60 and HCon60) and five repetitions of eccentric knee extension at 60 deg/s (HEcc60) with the highest peak torque value recorded, in addition to the functional hamstring-to-quadriceps ratio (H:Q = HEcc60:QCon60). A minimum of 60 s of rest was provided between each contraction mode. Before each isokinetic test, participants were instructed as to the mode and procedure of the specific test and provided with appropriate practice repetitions. During testing, vigorous verbal encouragement was provided by the assessors.

Participants performed the Nordic hamstring exercise test on the NordBord (Vald Performance, Albion, Australia) in a kneeling position, chest and hips extended, arms across their chest and ankles secured by individual ankle braces which are attached to uniaxial load cells. Players were instructed to lower their body as slowly as possible to their maximum depth or until they achieved a prone position. After catching their fall, each player pushed themselves back to the start position to minimize concentric knee flexor activity. Three trials were completed with a 30 s rest period between each repetition. In addition to absolute and relative force values, the percentage of body mass-expected eccentric strength was calculated (Body mass-expected eccentric strength (Nm) = 4 × BM (kg) + 26.1), as described by Buchheit et al. [36].

Eccentric hip adduction (ADD) and abduction (ABD) strength were evaluated with the athlete in a side-lying position using the break test method in accordance with previously published protocols [26,37]. Strength values were quantified using a hand-held dynamometer (PowerTrack II Commander, JTECH Medical, Midvale, UT, USA). Three repetitions were performed on each side with a 30 s rest period between trials, and raw force values (N) were reported.

#### 2.3.4. Statistical Analyses

Descriptive statistics (mean ± standard deviation (SD)) were calculated for each variable. Normality of the data was assessed using visual inspection and the Shapiro–Wilk test. A one-way analysis of variance (ANOVA) was used to examine for any positional differences in each physical quality, applying a Bonferroni post-hoc correction. Homogeneity of variance was tested via Levene’s test. All data were computed through Microsoft Excel^®^ 2010. The ANOVA and tests for normality and variance were processed using SPSS^®^ V.22 (IBM, Chicago, IL, USA) with the level of statistical significance set at alpha level *p* ≤ 0.05. Cohen’s *d* effect sizes were calculated to interpret the magnitude of between-group differences using the following classifications: standardized mean differences of 0.2, 0.5, and 0.8 for small, medium, and large effect sizes, respectively [38].

## 3. Results

### 3.1. Anthropometrics

Anthropometric characteristics for all players are provided in Table 1. Results indicated goalkeepers were taller than midfield players (*p* < 0.001; *d* = 1.27), and heavier (DEF *p* < 0.001, *d* = 1.31; MID *p* < 0.001, *d =* 1.60; STR *p* < 0.01, *d* = 0.86) with a higher BMI (DEF *p* < 0.001, *d =* 1.07; MID *p* < 0.001, 1.09; STR *p* < 0.05, *d =* 0.71) compared to all outfield positions. In addition, strikers were heavier than midfielders (*p* < 0.05, *d =* 0.51). No other meaningful between-position differences were evident.

### 3.2. Range of Motion

No significant positional differences for any range of motion tests were observed (Table 2). A pattern was present where strikers displayed heightened hip IR range of motion with medium effect sizes when compared to defenders (*d* = 0.54) and goalkeepers (*d* = 0.73) respectively, whereas ankle dorsiflexion was greatest in goalkeepers although the strength of these between-group effects was small (*d* < 0.38). Goalkeepers also showed the greatest hip flexibility during the BKFO test, but all effect sizes were small (*d* < 0.25). Variable results were observed between positions in hamstring range of motion.

### 3.3. Jump Performance

No significant differences in jump height were apparent between positions for either bilateral or unilateral countermovement jumps with small effect sizes (*d* < 0.41) (Table 3). For the 10 s Hop test, there were no positional differences in average jump height; however, the average RSI score was higher for midfielders than goalkeepers when using the non-dominant leg and this corresponded to a medium effect size (*d* = 0.69).

### 3.4. Strength

In terms of absolute strength, goalkeepers were stronger than midfielders for QCon60 in both limbs (*p* < 0.01, *d* = 0.86) and for HCon60 on their dominant side (*p* < 0.05, *d* = 0.78) (Table 4). Goalkeepers also demonstrated higher strength values than strikers for QCon60 in the dominant leg (*p* < 0.01, *d* = 0.87) and defenders for the Groin squeeze test (*p* < 0.05, *d* = 0.77). Strikers displayed a greater functional H:Q ratio with a strong effect size when compared to the dominant side against goalkeepers (*p* < 0.05, *d* = 0.88). Normalized to body mass, defenders were stronger than all other positions during concentric measures of quadriceps and hamstrings, with a significant difference observed in the dominant leg compared to strikers for QCon60 (*p* < 0.01, *d* = 0.53) (Table 5). Defenders also performed at a higher percentage of body mass-expected eccentric strength than goalkeepers when examining the dominant leg in the Nordic hamstring exercise corresponding to a strong effect size (*p* < 0.05, *d* = 0.80). In relation to hip strength, defenders demonstrated higher values in both legs for eccentric hip abduction than goalkeepers with a strong effect size (*p* < 0.05, *d* = 0.85–0.87). Midfielders were relatively stronger compared to goalkeepers for both eccentric hip abduction (dominant *p* < 0.01, *d* = 1.0; non-dominant *p* < 0.05, *d* = 0.70) and adduction (*p* < 0.05, *d* = 0.65–0.70), and compared to strikers for dominant leg hip abduction (*p* < 0.05, *d* = 0.54) and hip adduction in both legs (*p* < 0.05, *d* = 0.52–0.57).

## 4. Discussion

Results of this study indicate variation in player position across the different range of tests for anthropometric characteristics, range of motion, jump performance, and strength, suggesting that not all physical qualities are inter-related. Significant differences were shown in body mass and stature, with goalkeepers being generally taller and heavier than other positions. Such differences resulted in higher absolute strength, whereas relative strength measures appeared to favor defenders. Conversely, jump height and range of motion were similar between positions.

### 4.1. Anthropometrics

Goalkeepers were significantly taller than midfielders and heavier than all other positions which is consistent with other elite football leagues [4,6,7,9,14,39,40], and most likely reflects the position-specific demands. Goalkeepers are often required to contest for the ball in the air, which favors players of greater stature and mass, and a taller goalkeeper will also have a longer reach which is advantageous for covering larger parts of the goal [4,14]. Midfielders were the smallest and lightest of all positions, which is supported by previous studies from other leagues [4,6,7,9,14,39,40]. Again, this could reflect positional demands with midfield players covering significantly greater total distances than defenders and strikers [1], with coaches likely placing greater emphasis on technical and tactical abilities, and the capacity to cover large distances for players in these positions [4,6].

The stature of a QSL player is similar to professional players in the United Arab Emirates (175.1 cm) [21] and soccer players in Saudi Arabia (177.2 cm) [22], Kuwait (172.7 to 173.4 cm) [23,24] and Bahrain (172.0 cm) [25], but shorter compared to the top divisions in England (181 cm) [15], Spain (180.6 cm) [41], Norway (182.9 cm) [7], Iceland (181 cm) [9], Belgium (178.4 cm) [42], Poland (181cm) [40] and Serbia (181.8 cm) [8]. A similar trend in body mass was also shown for these respective league comparisons but not in body mass index (BMI) with values of 23.1 kg/cm² consistent with other geographical locations [7,9,21].

### 4.2. Range of Motion

No meaningful positional differences were identified for any of the ROM tests. This finding is in accordance with research conducted in South African professional players [14]. Conversely, in the Icelandic league, goalkeepers demonstrated greater hip flexion and extension than outfield positions [9].

Overall values for hip range of motion in the QSL were comparable for internal rotation to a cohort of players competing in elite Danish soccer [43], and exceeded reported values in an Australian cohort for the BKFO test [10]. Hamstring flexibility was symmetrical with values around 88° reported during the passive knee extension test. Similarly, ankle dorsiflexion in QSL players was symmetrical with values of 10 cm reported during the weight-bearing lunge test. Available research to report ankle range of motion in elite soccer players is sparse and has used passive techniques [13,44]. Measurement of ankle dorsiflexion using weight-bearing and non-weight bearing tasks has identified significant differences and only moderate correlations between these two assessment modes [45]. Cumulatively, the range of motion displayed in relevant musculature by QSL players in the current study appears comparable; however, caution should be applied as there is a paucity of data in these cohorts, with significant variability in the test protocols utilized across studies [26].

### 4.3. Jump Performance

No significant positional differences were evident for CMJ or SLCMJ height, or for average 10 s Hop height. The absence of positional differences for jump performance is reiterated by previous findings in professional players in Iceland [9], South Africa [14], and Division 1 NCAA college players in the USA [46]. In contrast, forwards outperformed midfielders and full-backs in Belgium [4], and in both Croatia and Norway, midfielders showed significantly lower jump height values than the other positions [6,7]. Furthermore, goalkeepers jumped significantly higher than the outfield positions in Croatia [6]. Intuitively, certain positions would benefit from the ability to execute higher vertical jumps, especially those that regularly contend heading duels in dangerous areas in front of the goal, such as goalkeepers, defenders, and strikers. Strikers are also often considered the most explosive of all positions and although not significantly different, they displayed the greatest jump heights in the current study. Greater reactive strength shown during the 10 s Hop test in midfielders is likely due to lower body mass and decreased force attenuation demands; thus, further development of relative strength and fast stretch shortening cycle function through the use of plyometrics is warranted in the other positional categories.

Direct comparisons with previous literature are difficult, as a variety of jumping techniques and assessment tools have been used. For example, a range of jump height calculation methods are available which provide different results [47] and the inclusion of an arm swing during countermovement jumps has also been shown to augment jump height due to an increase in lower extremity work performed [48]. The present study used a countermovement jump with hands on hips and derived jump height from flight time measured using dual portable force plates. This method has been shown to be accurate within 1 cm of the criterion measure, a laboratory ground fixed force platform; whereas, the validity of contact mats and Vertec devices has been questioned [49]. The average vertical jump height of QSL players was lower than elite soccer players in England (40 cm) [15], Spain (44.9 cm) [41], Iceland (39.4 cm) [9], and Belgium (41 to 46 cm) [4], although these studies utilized a contact mat. CMJ values of 39 cm were reported in Norway [7], 49.9 cm in Serbia [8], and between 44.2 to 48.5 cm in Croatia depending on playing position when using force plates [6]. Single-leg vertical jump height was also measured in the current study with average performances of 17.3 cm. To the knowledge of the authors, only one study has reported unilateral scores in this cohort with values of 25.0 and 25.2 cm for the dominant and non-dominant leg, respectively [50].

### 4.4. Strength

In this study, relative isokinetic knee extension and flexion strength were higher in defenders, with a more variable pattern evident in hip strength. Again, the literature available to examine positional differences in strength for elite football cohorts is limited, and methodological differences complicate any direct comparisons [40]. Early research describing the physiological characteristics of elite English soccer players showed similar extensor torque and extensor-flexor ratios between positions when corrected for body mass [51]. Similar to the findings of the current study, midfielders produced the lowest absolute peak torque while defenders demonstrated the greatest relative peak torque values. More recent data from Brazilian professional players reported greater quadriceps and hamstring strength in goalkeepers than other positions [52]. However, these values were not normalized; thus, the reported data does not allow a fair comparison due to the influence of differences in body mass on torque output. Supporting this influence of body mass on torque output, Sliwowski, Grygorowicz, Hojszyk, and Jadczak [40] investigated position-specific isokinetic strength profiles in elite Polish soccer players, and their results indicated that relative strength was generally lower for goalkeepers and midfielders when compared to other playing positions. Furthermore, in a Swedish study on senior soccer teams, greater absolute knee extensor torque values displayed in goalkeepers and defenders were no longer evident when corrected for body mass [53]. Positional differences may be reflective of the demands of match play with defenders and strikers expected to perform more short duration high-intensity actions than midfield players [40,54], although our results do not support this notion. Defenders were consistently the strongest in relative terms, with midfielders recording the next highest values as well as displaying significantly greater eccentric hip strength than goalkeepers and strikers. 

The findings of this study indicate that overall isokinetic knee extension and flexion strength profiles of QSL players are lower than other elite football populations, and both Greek divisional football [55], English Premier League [15], and elite and amateur French soccer players [56] have demonstrated greater absolute peak torque values at 60 deg/s. Few studies have reported relative values which in part may explain the apparent reduction in absolute strength due to lower body mass in the current cohort compared to other leagues, making direct comparisons with non-normalized data and subjects with a wide range of body sizes difficult. Higher levels of relative quadriceps (dominant: 3.52 ± 0.65 Nm/kg; non-dominant: 3.49 ± 0.69 Nm/kg) and hamstring (dominant: 2.17 ± 0.55 Nm/kg; non-dominant: 2.03 ± 0.41 Nm/kg) peak torque at 60 deg/s have been reported in elite Premier League footballers [15], and similarly, relative torque for quadriceps (3.2 Nm/kg) and hamstrings (dominant: 1.9 Nm/kg; non-dominant: 1.8) were greater in Polish elite players [40] than what was seen in the QSL. Functional H:Q ratio (0.76) was also lower than previous research in a sample of Greek football league players (1.1) [55], but greater than Brazilian [52] and elite, sub-elite, and amateur French soccer players (0.55–0.67) [56]. Previous recommendations for a healthy ratio have been suggested to range from 0.5 to 0.8 [57], which suggests that QSL values are appropriate; however, as strength levels are lower than other leagues, targeted strength training programs to develop both quadriceps and hamstring strength are warranted.

### 4.5. Limitations

A major strength of this study is the relatively large sample size compared to similar profiling studies, yet some limitations must be acknowledged. The cross-sectional design only provides a single time-point to capture players’ performance, which is compounded by the potential for inter-season variation that has previously been demonstrated for the isokinetic and Nordic hamstring tests used in this study [58]. In addition, sports specific assessments to characterize sprint and change of direction speed and aerobic capacity were not included due to the nature of the testing, with a focus on pre-participation screening as opposed to on-pitch physical performance.

## 5. Conclusions

The present study provides position-specific normative values for players who compete in the Qatar Stars League during commonly used musculoskeletal and performance tests. Positional differences were apparent for anthropometric and strength measures, while jump performance and range of motion were similar between positions. Compared to mean values from other professional leagues, players appear to be shorter and lighter and are also inferior in terms of jumping and strength tests, although direct comparisons must be interpreted with caution given the wide range of methodologies applied in previous literature. The results of this study highlight the potential importance of training and rehabilitation programs to be tailored to the specific playing position and league in which the athletes compete.

## Figures and Tables

**Table 1 sports-07-00009-t001:** Demographics and anthropometric characteristics for each playing position, with averages and ranges for all positions combined. Letters in italic indicate significantly greater values (*p* < 0.05) compared to goalkeepers (g), defenders (d), midfielders (m) and strikers (s).

Variable	Goalkeepers		Defenders	Midfielders	Strikers		Overall	Range
	Mean ± SD		Mean ± SD	Mean ± SD	Mean ± SD		Mean ± SD	Min–Max
n	19		60	78	38		195	
Age (years)	23.5 ± 4.8		24.7 ± 5.1	24.2 ± 4.5	24.9 ± 4.7		24.4 ± 4.7	17.3–36.5
Height (cm)	180.1 ± 4.0	*m*	176.4 ± 7.0	173.6 ± 6.0	176.9 ± 6.9		175.7 ± 6.6	160.0–193.0
Body mass (kg)	81.4 ± 7.9	*dms*	71.0 ± 8.0	68.6 ± 8.1	73.4 ± 10.6	*m*	71.5 ± 9.3	49.0–97.0
BMI (kg/cm^2^)	25.1 ± 2.3	*dms*	22.8 ± 2.0	22.7 ± 2.1	23.4 ± 2.5		23.1 ± 2.3	17.6–28.7

**Table 2 sports-07-00009-t002:** Range of motion for each playing position, with averages and ranges for all positions combined. IR = Internal rotation, BKFO = Bent knee fall out test, PKET = Passive knee extension test.

Variable	Goalkeepers	Defenders	Midfielders	Strikers	Overall	Range
	Mean ± SD	Mean ± SD	Mean ± SD	Mean ± SD	Mean ± SD	Min − Max
Hip IR (deg)						
Dominant	33.5 ± 5.8	34.0 ± 8.2	35.1 ± 8.1	37.9 ± 6.2	35.2 ± 7.7	16.0–59.5
Non-dominant	34.2 ± 6.3	34.6 ± 8.5	35.4 ± 7.8	37.5 ± 7.9	35.4 ± 7.9	18.5–58.0
BFKO (cm)						
Dominant	13.2 ± 4.7	14.1 ± 3.8	13.8 ± 3.9	14.6 ± 4.4	14.0 ± 4.0	4.0–24.0
Non-dominant	13.0 ± 3.7	14.5 ± 4.4	13.4 ± 3.8	14.4 ± 5.0	13.9 ± 4.2	4.0–26.0
PKET (deg)						
Dominant	87.9 ± 11.7	88.6 ± 9.7	87.5 ± 9.4	85.7 ± 10.1	87.5 ± 9.9	67.9–115.9
Non-dominant	86.4 ± 10.8	86.8 ± 9.8	87.7 ± 10.3	86.7 ± 10.7	87.1 ± 10.2	64.7–118.7
Dorsiflexion (cm)						
Dominant	11.1 ± 3.7	10.3 ± 3.2	10.0 ± 3.0	9.8 ± 3.2	10.2 ± 3.2	2.0–19.0
Non-dominant	10.9 ± 3.2	10.1 ± 3.1	10.2 ± 3.1	9.8 ± 3.2	10.1 ± 3.1	2.0–20.5

**Table 3 sports-07-00009-t003:** Jump performance for each playing position, with averages and ranges for all positions combined. CMJ = Countermovement jump, SLCMJ = Single-leg countermovement jump, RSI = Reactive strength index. Letters in italic indicate significantly greater values (*p* < 0.05) compared to goalkeepers (g), defenders (d), midfielders (m) and strikers (s).

Variable	Goalkeepers	Defenders	Midfielders		Strikers	Overall	Range
	Mean ± SD	Mean ± SD	Mean ± SD		Mean ± SD	Mean ± SD	Min − Max
CMJ (cm)	35.1 ± 5.0	34.8 ± 4.9	34.1 ± 4.4		35.8 ± 3.8	34.7 ± 4.5	23.5–47.3
SLCMJ (cm)							
Dominant	17.5 ± 2.6	17.4 ± 3.2	17.1 ± 3.1		17.6 ± 2.9	17.3 ± 3.0	8.8–25.5
Non-dominant	17.2 ± 2.9	17.3 ± 3.1	17.4 ± 2.6		17.2 ± 3.8	17.3 ± 3.0	5.9–26.6
10 s Hop Average (cm)							
Dominant	11.6 ± 2.7	13.3 ± 3.1	12.9 ± 3.1		13.2 ± 2.6	13.0 ± 3.0	4.9–21.8
Non-dominant	11.7 ± 3.5	13.1 ± 2.8	13.5 ± 2.8		12.9 ± 3.4	13.1 ± 3.0	3.6–23.3
10 s Hop Average RSI							
Dominant	0.36 ± 0.10	0.40 ± 0.10	0.43 ± 0.11		0.41 ± 0.09	0.41 ± 0.11	0.11–0.70
Non-dominant	0.37 ± 0.12	0.40 ± 0.10	0.45 ± 0.11	*g*	0.40 ± 0.12	0.42 ± 0.11	0.12–0.74

**Table 4 sports-07-00009-t004:** Absolute strength values for each playing position, with averages and ranges for all positions combined. QCon60 = Concentric quadriceps at 60 deg/s, HCon60 = Concentric hamstring at 60 deg/s, HEcc60 = Eccentric hamstring at 60 deg/s, ABD = Abduction, ADD = Adduction. Functional H:Q ratio = HEcc60:QCon60. Letters in italic indicate significantly greater values (*p* < 0.05) compared to goalkeepers (g), defenders (d), midfielders (m) and strikers (s).

Variable	Goalkeepers		Defenders	Midfielders	Strikers		Overall	Range
	Mean ± SD		Mean ± SD	Mean ± SD	Mean ± SD		Mean ± SD	Min − Max
QCon60 (Nm)								
Dominant	244.0 ± 44.9	*ms*	221.6 ± 41.1	207.5 ± 39.4	204.7 ± 48.1		214.9 ± 43.6	102.0–358.0
Non-dominant	244.4 ± 45.3	*m*	224.5 ± 44.3	207.8 ± 39.9	215.8 ± 46.9		218.1 ± 44.3	110.0–333.0
HCon60 (Nm)								
Dominant	128.7 ± 21.9	*m*	120.7 ± 23.6	111.6 ± 21.7	123.5 ± 32.4		118.4 ± 25.2	66.0–189.0
Non-dominant	125.7 ± 27.4		118.3 ± 21.9	108.0 ± 24.4	120.3 ± 35.6		115.3 ± 27.1	43.0–199.0
HEcc60 (Nm)								
Dominant	168.1 ± 27.8		161.4 ± 33.1	156.7 ± 32.5	166.6 ± 44.4		161.2 ± 34.9	92.0–278.0
Non-dominant	160.7 ± 30.4		159.8 ± 29.2	150.8 ± 32.6	166.3 ± 45.5		157.6 ± 34.6	70.0–269.0
Functional H:Q ratio								
Dominant	0.70 ± 0.12		0.74 ± 0.16	0.77 ± 0.15	0.82 ± 0.15	*g*	0.76 ± 0.15	0.39–1.33
Non-dominant	0.67 ± 0.11		0.73 ± 0.15	0.74 ± 0.15	0.78 ± 0.17		0.74 ± 0.15	0.37–1.30
NordBord Peak (Nm)								
Dominant	362 ± 79		337 ± 76	344 ± 79	369 ± 92		358 ± 81	156–677
Non-dominant	340 ± 93		335 ± 72	325 ± 71	329 ± 75		330 ± 74	122–535
Ecc. Hip ABD (Nm)								
Dominant	236 ± 46		232 ± 41	230 ± 40	224 ± 52		230 ± 43	123–349
Non-dominant	237 ± 47		232 ± 38	226 ± 46	222 ± 45		228 ± 43	115–330
Ecc. Hip ADD (Nm)								
Dominant	263 ± 59		250 ± 45	249 ± 47	241 ± 60		249 ± 50	140–416
Non-dominant	260 ± 61		244 ± 37	246 ± 49	240 ± 55		246 ± 48	132–403
Groin Squeeze (N)	289.9 ± 62.7	*d*	255.0 ± 50.6	269.8 ± 54.1	263.7 ± 69.3		266.9 ± 58.0	127.0–457.0

**Table 5 sports-07-00009-t005:** Strength values for each playing position relative to body mass, with averages and ranges for all positions combined. Body mass (BM)-expected eccentric strength is calculated with the equation described by Buchheit et al. (2016) [36] (Eccentric strength (Nm) = 4 x BM(kg) + 26.1). QCon60 = Concentric quadriceps at 60 deg/s, HCon60 = Concentric hamstring at 60 deg/s, HEcc60 = Eccentric hamstring at 60 deg/s, ABD = Abduction, ADD = Adduction. Functional H:Q ratio = HEcc60:QCon60. Letters in italic indicate significantly greater values (*p* < 0.05) compared to goalkeepers (g), defenders (d), midfielders (m) and strikers (s).

Variable	Goalkeepers	Defenders		Midfielders		Strikers	Overall	Range
	Mean ± SD	Mean ± SD		Mean ± SD		Mean ± SD	Mean ± SD	Min − Max
QCon60 (Nm/kg)								
Dominant	3.00 ± 0.51	3.13 ± 0.50	*s*	3.03 ± 0.46		2.80 ± 0.55	3.01 ± 0.50	1.49–4.22
Non-dominant	3.00 ± 0.46	3.17 ± 0.58		3.03 ± 0.43		2.94 ± 0.45	3.05 ± 0.49	1.55–4.34
HCon60 (Nm/kg)								
Dominant	1.59 ± 0.27	1.71 ± 0.30		1.63 ± 0.26		1.67 ± 0.32	1.66 ± 0.29	0.93–2.63
Non-dominant	1.54 ± 0.27	1.67 ± 0.26		1.57 ± 0.27		1.63 ± 0.38	1.61 ± 0.29	0.79–2.52
HEcc60 (Nm/kg)								
Dominant	2.07 ± 0.33	2.28 ± 0.43		2.29 ± 0.41		2.27 ± 0.50	2.26 ± 0.43	1.27–3.58
Non-dominant	1.98 ± 0.35	2.26 ± 0.38		2.20 ± 0.41		2.27 ± 0.53	2.21 ± 0.43	1.16–3.41
NordBord Peak (Nm/kg)								
Dominant	4.42 ± 0.76	5.20 ± 1.03	*g*	5.03 ± 1.09		5.04 ± 1.06	5.02 ± 1.05	2.36–8.26
Non-dominant	4.14 ± 0.97	4.74 ± 0.97		4.75 ± 0.89		4.51 ± 0.93	4.64 ± 0.94	2.07–7.31
BM-Expected Ecc. Strength (%)								
Dominant	2.2 ± 17.9	18.8 ± 23.3	*g*	14.7 ± 24.6		15.5 ± 24.4	14.9 ± 23.9	–46.2–91.2
Non-dominant	-4.2 ± 22.6	8.5 ± 22.0		8.3 ± 20.2		3.2 ± 21.0	6.1 ± 21.4	–52.0–67.5
Ecc. Hip ABD (Nm/kg)								
Dominant	2.89 ± 0.40	3.27 ± 0.47	*g*	3.37 ± 0.55	*gs*	3.06 ± 0.59	3.23 ± 0.54	1.95–4.60
Non-dominant	2.90 ± 0.42	3.28 ± 0.47	*g*	3.30 ± 0.59	*g*	3.03 ± 0.49	3.20 ± 0.54	1.80–4.83
Ecc. Hip ADD (Nm/kg)								
Dominant	3.21 ± 0.57	3.55 ± 0.66		3.65 ± 0.65	*gs*	3.28 ± 0.64	3.50 ± 0.66	1.92–5.57
Non-dominant	3.18 ± 0.63	3.47 ± 0.56		3.60 ± 0.67	*gs*	3.27 ± 0.60	3.46 ± 0.63	2.00–5.15
Groin Squeeze (N)	3.68 ± 0.72	3.63 ± 0.79		3.96 ± 0.78		3.62 ± 0.94	3.77 ± 0.82	1.49–6.77

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
