# Peer review of "Examination of Physical Characteristics and Positional Differences in Professional Soccer Players in Qatar"

_sports, 2018, doi:10.3390/sports7010009_

Round 1

Reviewer 1 Report

The manuscript provides an analysis of the physical characteristics in professional players competing in the Qatar Stars League. I would like to commend the authors on providing a well written and interesting analysis. Personally, I would debate the need for the test and some device used, but the manuscript provides relevant information about a soccer competition not researched a lot. The comments below are provided as a means to enhance the current manuscript.

General comments.

There is some good analysis about the situation in the introduction, but I believe the manuscript could be enhanced further by the inclusion of more detail related to the tests used to evaluate the physical characteristics in professional soccer players. This could be help understand better why the authors used these tests and no others. In the world of the soccer tests there are a lot of tests to stimate the soccer players profile characteristics.

The method is particularly well presented, with detailed information provided. However, the author has to be in consideration that Anthropometric information should be recorded using the skinfold reference not just the body mass, because the body mass data has been demonstrated not useful as relevant information. By other hand statistical analyses could be improved adding effect size information. Correlational test could be interesting to know possible associations between variables that could explain more the results obtained and give more reasons to individualize the train sessions in professional soccer players.

Results.

Adjust table 4 at the same page and add the effect size in all tables.

Discussion and Conclusions.

There is some good discussion and conclusion, with good references.

References:

Review reference 21 (p11-412).

Author Response

COMMENT:

The manuscript provides an analysis of the physical characteristics in professional players competing in the Qatar Stars League. I would like to commend the authors on providing a well written and interesting analysis. Personally, I would debate the need for the test and some device used, but the manuscript provides relevant information about a soccer competition not researched a lot. The comments below are provided as a means to enhance the current manuscript.

RESPONSE:

Thank you for both your time and the constructive feedback provided on the manuscript. Your expertise and suggested comments are welcomed and have allowed us to further improve the quality of the manuscript. We agree that the tests included in the screening process can be debated, and thus should be continuously evaluated based on their relevance and how they reflect the purpose of the assessment. Nonetheless, these tests are used frequently both in the applied and research settings and we hope that practitioners can benefit from this analyses and interpret them in light of the similarities and differences between our population sample and their own context.

GENERAL COMMENTS:

There is some good analysis about the situation in the introduction, but I believe the manuscript could be enhanced further by the inclusion of more detail related to the tests used to evaluate the physical characteristics in professional soccer players. This could be help understand better why the authors used these tests and no others. In the world of the soccer tests there are a lot of tests to stimate the soccer players profile characteristics.

The method is particularly well presented, with detailed information provided. However, the author has to be in consideration that Anthropometric information should be recorded using the skinfold reference not just the body mass, because the body mass data has been demonstrated not useful as relevant information. By other hand statistical analyses could be improved adding effect size information. Correlational test could be interesting to know possible associations between variables that could explain more the results obtained and give more reasons to individualize the train sessions in professional soccer players.

RESPONSE:

Unfortunately, given the nature of this screening process, field tests such as sprinting and change of direction assessments were not a practically viable option during this preseason medical evaluation. We agree that skinfold assessments could have added to the profiling in terms of body composition, however this was unfortunately not included due to time constraints of screening a large number of players and the high volume of tests already undertaken. We have carefully considered including further details to the methods used, however, this would greatly inflate the word count. In the current manuscript we refer to previously published work describing the methods, and we would suggest that this is sufficient for the format of the article. If the reviewer disagrees with our argumentation, we would suggest that an online supplement or appendix could be a better means for providing the reader with detailed descriptions.

Regarding the additional statistical analysis, we have now included effect sizes in the results section in accordance with your recommendations to give the reader a more detailed picture of the magnitude of these differences between playing positions. Given the already large amount of data presented in the article, we would suggest that correlation analyses would overcomplicate and distract from the main focus and aims of this study; to provide normative data for an under researched population and report the positional differences in test performance. It would, however, be an interesting topic for a follow-up piece with a more methodological approach.

COMMENT (RESULTS):

Adjust table 4 at the same page and add the effect size in all tables.

RESPONSE:

We have now adjusted the table to fit on one page as requested and agree that this makes the tables easier for the reader to understand. 

COMMENT (DISCUSSION AND CONCLUSIONS):

There is some good discussion and conclusion, with good references.

RESPONSE:

Thank you for your positive feedback.

COMMENT (REFERENCES):

Review reference 21 (p11-412).

RESPONSE:

The reference has now been updated.

Reviewer 2 Report

This study was novel and very well done. There are some limitations with isokinetic strength testing however the large sample helps with this tremendously. Studies like this are incredibly important in our field as they help us better understand our athletes. It was so well written I have very few comments (mention the specific tests in the abstract and better describe the QSL for readers who aren't familiar with the league). Kudos to the authors!

abstract: specific tests should be mentioned in the abstract (may need to remove some to allow for it to not be big too long)

86: a sentence describing the QSL would be helpful for those unfamiliar with the league (e.g. is it best players in the country?) 

239 - 245: really good comparison with other countries 

303: god mention of limitation for comparing strength assessment 

Author Response

GENERAL COMMENTS:

This study was novel and very well done. There are some limitations with isokinetic strength testing however the large sample helps with this tremendously. Studies like this are incredibly important in our field as they help us better understand our athletes. It was so well written I have very few comments (mention the specific tests in the abstract and better describe the QSL for readers who aren't familiar with the league). Kudos to the authors!

RESPONSE:

Thank you for your positive response!

COMMENT (ABSTRACT):

specific tests should be mentioned in the abstract (may need to remove some to allow for it to not be big too long)

RESPONSE:

We agree that providing some further details of the tests in the abstract will be useful for the readers to gain a greater appreciation of the screening content. We have now expanded our original generic descriptions to provide more insight for the readers but have also had to do so in the constraints of the word count of the abstract. We hope you agree that our revised abstract now provides sufficient details for the readers to ascertain what was asked of the players and further details of each test are then provided in the methods where we are able to expand our description of each mode of testing used. Thank you for your suggestion here.

COMMENT (METHODS):

86: a sentence describing the QSL would be helpful for those unfamiliar with the league (e.g. is it best players in the country?) 

RESPONSE:

We agree this would be useful to provide greater context for the readers and have included a short description of the QSL as requested.

COMMENT (DISCUSSION):

239 - 245: really good comparison with other countries 

RESPONSE:

Thank you for your positive feedback.

COMMENT (DISCUSSION):

303: good mention of limitation for comparing strength assessment 

RESPONSE:

Thank you.

Reviewer 3 Report

This paper provides position specific normative values for soccer player. In general such data has a high value, especially because the number of participants is rather high. The manuscript is well written and tightly organized. 

l102. Please include a statement about nutrition / water state of the athletes. 

Where the tests in randomized orders?

Table 4: I suggest to round up the numbers for Nord Peak, Ecc Hip ABD /ADD to  362 pm 79

Author Response

GENERAL COMMENTS:

This paper provides position specific normative values for soccer player. In general, such data has a high value, especially because the number of participants is rather high. The manuscript is well written and tightly organized. 

RESPONSE:

Thank you for taking the time to review our work and the positive response!

COMMENT (METHODS):

l102. Please include a statement about nutrition / water state of the athletes. 

Where the tests in randomized orders?

RESPONSE:

We have now specified the standardization of test order and included details of participant instructions regarding nutritional and fluid intake.

COMMENT (RESULTS):

Table 4: I suggest to round up the numbers for Nord Peak, Ecc Hip ABD /ADD to  362 pm 79

RESPONSE:

Changed as suggested.